# CD301b+ Macrophages as Potential Target to Improve Orthodontic Treatment under Mild Inflammation

**DOI:** 10.3390/cells12010135

**Published:** 2022-12-29

**Authors:** Nan Wang, Qin Zhao, Zijian Gong, Liangliang Fu, Jiaojiao Li, Li Hu

**Affiliations:** 1Department of Stomatology, Union Hospital, Tongji Medical College, Huazhong University of Science and Technology, Wuhan 430022, China; 2School of Stomatology, Tongji Medical College, Huazhong University of Science and Technology, Wuhan 430030, China; 3The State Key Laboratory Breeding Base of Basic Science of Stomatology (Hubei-MOST) & Key Laboratory of Oral Biomedicine Ministry of Education, School & Hospital of Stomatology, Wuhan University, Wuhan 430079, China

**Keywords:** macrophage, orthodontic, inflammation, bone regeneration, bone resorption, single-cell RNA-seq

## Abstract

Due to improvements of quality of life and the demand for aesthetics, more and more people are choosing orthodontic treatments, resulting in a surge in adult orthodontic patients in recent years. However, a large amount of clinical evidence shows that many orthodontic patients have mild periodontitis in the periodontal tissues, which affects the efficacy of the orthodontic treatment or aggravates the periodontal condition. Therefore, it is important to identify the key factors that affect orthodontic treatments in this inflammatory environment. The aim of this study was to investigate the role of macrophages in orthodontic treatments under inflammatory environments. By analyzing the functional groups of macrophages in the orthodontic rat model of periodontitis, we found that macrophages with high expression levels of CD301b could improve the periodontal microenvironment and improve the efficiency of the orthodontic tooth movement. CD301b+ macrophages transplanted into the model can promote osteogenesis around orthodontic moving teeth, improve bone remodeling during orthodontic treatment, and accelerate orthodontic tooth movement. Considered together, these results suggest that CD301b+ macrophages may play an active role in orthodontic treatments in inflammatory environments and may serve as potential regulatory targets.

## 1. Introduction

Malocclusion, along with periodontal disease and caries, is ranked as one of the three major diseases of the oral cavity by the World Health Organization (WHO). Orthodontic treatments can correct dental and jaw misalignments and improve the facial or dental aesthetics and masticatory functions, owing to the plasticity of the jaw [1]. Improvements in the standard of living and social civilization have led to a progressive increase in the demand for aesthetics. The number of patients requesting orthodontic treatments and the proportion of adults among those undergoing orthodontic treatments are steadily increasing [2,3]. The complexity of the oral environment dictates that orthodontic treatments are bound to be riskier and more challenging in adults, particularly due to the increase in the incidence of periodontal disease with age.

According to the WHO Oral Health Database, approximately 90% of adults have some form of periodontal issue [4], ranging from localized mild gingivitis to generalized severe periodontitis [5]. Orthodontic treatments are considered susceptibility factors for periodontal disease because orthodontic appliances can hinder the daily cleaning of the mouth [6]. The periodontium continues to remain in a slightly inflamed environment during orthodontic treatment. Hence, maintaining the efficiency of orthodontic tooth movement (OTM) within a mildly inflammatory environment and ensuring a balanced remodeling of the bone are of the utmost importance during orthodontic treatment.

In recent years, scientists have become increasingly interested in the possibility of replacing damaged cells in living organisms [7]. Researchers are trying to understand the regenerative capacity of these cell populations and the potential use of these cells for immunotherapy or the treatment of various diseases, hoping to improve the immune environment and cure diseases through certain cell populations [8]. Macrophages are a strategically located, highly plastic, and heterogeneous cell population that can rapidly change their functions in response to signals in the local microenvironment [9]. They play an important role in regulating inflammation and bone regeneration and in mediating innate immune responses [10,11,12,13]. Furthermore, as osteoclast precursor cells, macrophages play a crucial role in bone remodeling during tooth movement [14]. Several studies have focused on the two extreme phenotypes of macrophages: M1 and M2 polarization [15,16]. An imbalance in the macrophage phenotype disturbs the bone remodeling process in vivo, which may lead to challenges in enhancing the efficiency of the orthodontic treatment under mildly inflammatory conditions. However, a recent single-cell RNA sequencing (ScRNAseq) study has shown that the typical M1 and M2 phenotypes in vitro are inconsistent with the heterogeneity of the macrophages in vivo [17]. In this regard, we wanted to target a subpopulation of phenotypically and functionally defined macrophages for use in orthodontic models.

We focused on a subpopulation of macrophages with tissue-regeneration-promoting functions and high expression levels of CD301b protein [18]. These CD301b+ macrophages overlap with the M1/M2 polarized classification and perform significant immunomodulatory and regenerative functions [19,20]. Therefore, these cells may play a more comprehensive role than the traditional M2-like macrophages by being involved in both inflammatory and bone-regenerative repair processes [21,22]. Previous studies have demonstrated that CD301b+ macrophages are necessary not only for tissue regeneration [20], but also for bone defect repair and regeneration [23,24]. Therefore, it is suggested that these cells may be considered as targets for orthodontic treatment in the inflammatory environment. However, the specific mechanism remains unclear.

In this study, we established an orthodontic animal model in normal and periodontal inflammatory microenvironments to compare the efficiency, bone remodeling, and metabolic levels of orthodontic tooth movement (OTM) [25,26]. The results showed that the inflammatory conditions hindered the effectiveness of the orthodontic treatment. Among these, the macrophages appear to be involved in key regulatory roles. Our further experiments verified that the recruitment and activation of CD301b+ macrophage subsets in the microenvironment has benefits for bone remodeling and can alleviate the adverse effects of periodontitis. CD301b+ macrophages are a promising target in orthodontic therapy.

## 2. Materials and Methods

### 2.1. Data Collection and Processing of Single-Cell Transcriptome Sequencing

Publicly available scRNAseq datasets were obtained from the Gene Expression Omnibus DataSets (GSM5640072 and GSM5640073). The dimensionality reduction in the data was performed using the Seurat function. For quality control, cells with a percentage of mitochondrial genes below 10% and between 200 and 5000 genes detected were retained. The principal component analysis (PCA) was performed based on the 2000 top variable genes. For the FindClusters parameter, the resolution value was set to 0.5. Clusters were annotated based on the MouseRNAseqData dataset. We used the “FindAllMarkers()” function to obtain differentially expressed genes (DEGs).

### 2.2. Establishment of the Rat Model

The animal experiments were conducted in accordance with the policy of the Ethics Committee for Animal Research, School and Hospital of Stomatology, Wuhan University, China. The Ethics Committee for Animal Use approved the study under protocol number S0792203014. Male SD rats, weighing about 230–250 g, were used for the in vivo experiments after being acclimated to the SPF environment for one week. All animal experiments were approved by the Ethics Committee of Wuhan University. Establishment of the rat periodontitis model: After the rats were anesthetized with isoflurane, a groove of about 0.2 mm was ground on the palatal enamel of the maxillary first molar of the rat with a turbine. We used the needle holders to pass the 3-0 silk thread through the junction of the first molar and the second molar to the level of the gingival papilla, then wrapped it around the first molar and tied a knot to fix it. Establishment of the rat orthodontic model: After the rats were anesthetized with isoflurane, a groove with a depth of about 0.5 mm was ground in the distal part of the maxillary incisors of the rats with a turbine, and a 0.2 mm ligation wire was used to surround the incisors. We used a 0.2 mm ligature wire to pass under the abutment area between the first molar and the second molar and circle the first molar. We connected the ligature wire around the first molar to the ligature wire around the incisor with a tension spring and adjusted the tension to 50 g. After 12 h of postoperative fasting, food was taken and fed as usual.

### 2.3. Micro-CT Scans and Analysis

Bone tissue changes near the maxillary first molars were assessed using a micro-CT imaging system (Skyscan 1176). Scans were performed at 90 kV and 278 μA, the filter was Cu + Al, and the thickness of each layer was 9 μm in high-resolution mode. The tooth travel distance was measured using the DataViewer program. The CTAn program was used to calculate bone-tissue-related indicators, including the bone volume/total volume (BV/TV), trabecular bone number (Tb.N), trabecular bone thickness (Tb.Th), and trabecular bone separation (Tb.Sp). The circle around the root of the tooth was defined as the region of interest (ROI). The distance of the tooth movement was defined as the line between the distal most convex point of the first molar crown and the proximal most convex point of the first molar crown.

### 2.4. Histological Staining

The maxillary bone of each rat was taken as the specimen. The specimens after taking micro-CT images were placed in a decalcification solution containing 15% EDTA for 1 month, until the needle could penetrate the bone tissue without resistance. The specimens were dehydrated in a gradient manner, and the procedure was 70%, 80%, 90%, and 95% I and 95% II concentrations of alcohol for 2 h in sequence. The specimens were placed in n-butanol overnight and then dipped in wax and embedded the next day. The slice thickness range was chosen to be 5–8 μm. The specimen sections were placed in a 60 °C oven for 2 h before dewaxing. The sections were washed 3 times with PBS for 3 min each. For the H&E staining (Google Biotechnology, China), we used a hematoxylin staining solution for 10 s and then rinsed the sections with water for 1 min. Then, they were stained with eosin for 8 s and destained with 75% alcohol for 6 s. For the Masson staining (MXB Biotechnologies) and TRAP staining (MXB Biotechnologies), we followed the instructions in the manual.

For the immunohistochemical staining, the sections were first incubated with a drop of antigen retrieval solution at 37 °C for 30 min, then washed three times with PBS. Next, a drop of peroxidase blocking agent was added and they were incubated at 37 °C for 10 min and washed three times with PBS. Then, we added the serum and performed blocking at 37 °C for 1 h. We added the antibody solution (diluted 1:200 in PBS) and incubated the sections overnight at 4 °C. The next day, they were washed three times with PBS, then we added the biotin-labeled secondary antibody and them incubated at 37 °C for 1 h. We then washed them three times with PBS, added streptavidin–avidin peroxidase solution, and incubated them at 37 °C for 20 min. We then washed them three times with PBS, added a drop of freshly prepared DAB solution, and waited for 1 min. We then rinsed them with PBS and performed hematoxylin staining. For the IHC staining, the primary antibodies were IL-6 (Bio-Swamp, Shanghai, China), Col1a1 (Bio-Swamp, Shanghai, China), and TNF-α (Bio-Swamp, Shanghai, China).

For the immunofluorescence staining, the sections were first incubated with a drop of antigen retrieval solution at 37 °C for 30 min and then washed three times with PBS. Then, we added serum and blocked the sections at 37 °C for 1 h. Next, we added the antibody solution (diluted 1:200 in PBS) and incubated them overnight at 4 °C. The next day, we washed them three times with PBS, added the fluorescent secondary antibody (1:200 diluted with PBS), and incubated them at 37 °C for 1 h. They were then washed three times with PBS, mounted with DAPI-containing mounting medium, and stored at −20 °C.

### 2.5. Flow Cytometry

The periodontal tissue samples on the buccal and lingual sides of the maxillary first molars of each rat were excised with a scalpel and collected in a 24-well plate. We cut the tissue samples into pieces with scissors, added 1 mL of digestion solution (RPMI medium containing 20% collagenase type II and 20% collagenase type IV), and incubated them at 37 °C for 2 h. The tissue fluid was ground on a 70 μm filter under PBS washing, and the filtrate was collected and centrifuged at 2500 r for 5 min. The cells were transferred to a round-bottom 96-well plate and washed once with PBS, then the supernatant was spin-dried. We then added 50 µL of antibody solution (see Table 1), CD45 AP-cyanine7(Cy7) (Biolegend, San Diego, CA, USA), CD11b Pacific Blue (Biolegend, San Diego, CA, USA), F4/80 PE-cyanine7(Cy7) (Biolegend, San Diego, CA, USA), CD301b phycoerythrin (PE) (Biolegend, San Diego, CA, USA), and CD206 FITC (Biolegend, San Diego, CA, USA) and incubated them at 4 °C for 30 min. They were then centrifuged at 2500 r for 5 min and the supernatant was shaken off. After resuspending them in PBS and filtering them through a 70-micron filter, they were detected using a BD flow cytometer. The data analysis was performed using FlowJoc10.4 (Ashland, MA, USA) software.

### 2.6. Extraction and Culture of Primary Cells

Male SD rats of about 250 g were used and sacrificed after isoflurane anesthesia. The femur and tibia were cut off, and the attached muscle was stripped. After rinsing with PBS, we cut both ends of the backbone. The bone marrow was flushed out with a 20 mL syringe filled with the medium and collected into centrifuge tubes. We performed centrifugation at 1000 r for 5 min and discarded the supernatant. We added red blood cell lysate for 3 min, centrifuged the samples at 1000 r for 5 min, and discarded the supernatant. The cells were cultured in a 10 cm cell dish in DMEM high-glucose medium containing 100 U mL^−1^ penicillin, 100 μg mL^−1^ streptomycin, and 20% serum. Two days later, the supernatant in the cell dish was collected and the cells were taken after centrifugation to continue the culture. M-CSF (20 ng/mL) was added to obtain the BMDM. The adherent cells in the cell dish were further cultured to obtain BMSCs.

### 2.7. Real-Time Quantitative PCR

The P2 BMSCs were spread in a flexible six-well plate (FX-5000, Burlington, NC, USA), and after their cells were fused to 70%, the medium containing osteogenic induction solution (50 mL BMDM containing 10% FBS, 500 µL β-gp, 50 µL dexamethasone, 100 µL VC) was added and the cells were stretched using a device that stretches the cells (Flexcell, Burlington, NC, USA) for 48 h, then the culture was continued until the tenth day. The fluid was changed every three days.

We added 1 mL of TRIzol to each well of the six-well plate to extract the total cellular RNA. The corresponding cDNA was obtained using the Vazyme reverse transcription kit. The qRT-PCR system was loaded using Vazyme Taq Pro Universal SYBR qPCR Master Mix, and we detected the results using the CFX Connect system (Bio-Rad). Specific primers were purchased from Wuhan TianyiHuiyuan Biotechnology Co., Ltd. (Wuhan, China). All results were normalized based on the housekeeping gene glyceraldehyde-3-phosphate dehydrogenase (GAPDH). The primer sequences used in the qRT PCR process (5′ to 3′) are shown in Table 2:

### 2.8. CD301b+ Macrophage Transfusion

Rat primary BMDM samples were extracted and cultured for seven days to make them mature macrophages. Subsequently, the cultured mature BMDM samples were sorted by flow cytometry using ARIA BD. The CD301b+ macrophages and CD301b- macrophages were collected in 15 mL centrifuge tubes and resuspended in 100 µL of serum after centrifugation. The two types of cells were injected into the subgingival areas of orthodontically mobile teeth on both sides with well-established periodontitis orthodontic models. For each group of 10 rats, 10 µL was injected into each side. Approximately 200,000 cells were counted. This operation was repeated on days 0, 3, and 7 of the orthodontics for different cell reinfusion results.

### 2.9. ALP and Alizarin Red S Staining

The P2 BMSCs were plated in flexible six-well plates, then after the cells were fused to 70%, the osteogenic induction solution was added and cells were subjected to 48 h tension and the culture process was continued with the medium being changed every three days. After ten days of culture, the PBS samples were washed three times and stained with alizarin red and ALP staining reagents for calcified nodules, respectively, as well as for their ALP activity assay. Microscopic observations were then performed.

### 2.10. Enzyme-Linked Immunosorbent Assay (ELISA)

Orbital blood samples were collected from the rats in the experimental group and the blank control group. The blood samples were collected in 1.5 mL centrifuge tubes, and after standing for 2 h at 4 °C, the upper layer of serum was taken into new 1 mL centrifuge tubes and stored in a −80 °C refrigerator. The serum was then shipped on dry ice to the reagent manufacturer. The bone metabolic parameters were measured in each group using an ELISA kit (Bioswamp, Shanghai, China) according to the kit manufacturer’s protocol.

### 2.11. CCK-8 Proliferation Assay and Cell Scratch Assay

In total, 3000 cells were added to each well of a 96-well plate, then 10 μL of CCK-8 reagent (Dojindo, Kumamoto, Japan) was added to the 96-well plate at 12 h, 48 h, and 72 h of incubation, respectively, and the OD value was measured at 450 nm using a microplate reader (Thermo, Waltham, MA, USA) for 1 h at 37 °C.

The P2 BMSCs were incubated in six-well plates for 24 h and then scratched with a sterile 200 µL pipette tip. The cells were washed with PBS to remove any debris. Photographs were taken at 0, 6, 12, and 24 h post-wounding. The gap distance was assessed quantitatively using software such as ImageJ (National Institutes of Health).

### 2.12. Data Statistics and Analysis

All data are presented as means ± standard deviations. The *t*-test was used for the analysis of significance between the two groups, and an analysis of variance was used for three or more groups. A data analysis and image rendering were performed using GraphPad Prism 8.4 (Boston, MA, USA) software.

### 2.13. Availability of Data and Materials

All data generated or analyzed during this study are included in this published article and its Appendix A. All primary data supporting the results of this study can be obtained from the corresponding author upon reasonable request.

## 3. Results

### 3.1. Macrophages Play an Active Role in Orthodontics

To determine the changes in the macrophages during the orthodontics, a set of publicly available ScRNAseq data was analyzed. The sequencing data were derived from the alveolar bones of normal or orthodontically-treated mice. The data were subjected to a basic bioinformatics analysis followed by quality control and annotated into 11 clusters based on the MouseRNAseqData dataset (Figure 1A). The orthodontic group consisted of 2056 cells, and the control group consisted of 2845 cells. Next, we focused on the differences in the macrophages between the two groups (Figure 1B); 837 macrophages and 1352 macrophages were observed in the orthodontic and control groups, respectively. A dimensionality reduction was performed on the macrophage subsets and differentially expressed genes (DEGs) were identified using the FindAllMarkers() function (Figure 1C). Heatmaps and violin plots revealed the DEGs in the two groups (Figure 1C,D). The macrophages in the orthodontic group highly expressed genes that regulate bone metabolism, immune regulation, and angiogenesis, indicating that they could be involved in regulating the microenvironment during OTM and improving bone remodeling. Subsequently, Gene Ontology(GO) enrichment was performed based on the DEGs (Figure 1E,F). The macrophages in the orthodontic group demonstrated higher levels of bone remodeling, immune response, and other functions. In addition, unsupervised single-cell trajectories were constructed using Monocole, and significant differences in macrophage gene expression levels between orthodontic and control groups were observed (Figure 1G,H and Appendix A). These findings suggested that macrophages might improve the efficiency of OTM by regulating the microenvironment.

### 3.2. The Inflammatory Environment Affects the Efficiency of the OTM

In the rat model of periodontitis induced by ligation wire, mild inflammation such as gingival swelling, increased gingival crevicular fluid, and increased inflammatory factors in the gingival crevicular fluid appeared on the 10th day. Therefore, the rat OTM model was established by applying a mechanical device in the rat’s mouth for 10 days (Figure 2A). The rats were randomly divided into four groups as follows: control group (no treatment); orthodontic group (O group, wherein the maxillary first molars were orthodontically moved for ten days); orthodontic + periodontitis group (OP group, where the model was induced by tying a wire on the maxillary first molar and applying a tensile force to the maxillary first molar for 10 days); orthodontic after periodontitis treatment group (OPL group, in which, 10 days after the periodontitis model was established, the silk thread was removed for 3 days, then the O group model was established for 10 days). As such, the O group represents the pure orthodontic model, the OP group represents the orthodontic model with periodontitis, and the OPL group represents the orthodontic treatment after the periodontitis treatment. The numbers of F4/80+ and CD11b+ macrophages in the gingival soft tissues around orthodontic mobile teeth (first molars) were counted using flow cytometry, and the marker CD206 was detected (Figure 2B). The numbers of macrophages in the three experimental groups were lower than in the control group. The level of inflammation (inversely correlated with the expression of CD206) in the experimental group was the highest in the OPL group and the lowest in the O group. The micro-computed tomography (CT) imaging of the maxillae of the rats in the four groups revealed significant increases in the distance between the first and second molars in the three experimental groups following the application of the orthodontic force, indicating that the rat OTM model was successfully established (Figure 2C). The extent of movement of the first molars in the O group was significantly greater than in the other groups (Figure 2C,D). On the other hand, the bone volume/total volume (BV/TV) indices in the OP and OPL groups were lower than in the O group (Figure 2E); the distances of the cementoenamel junction–alveolar bone crest (CEJ-ABC) and the trabecular bone indices were not significantly different from those in the control group (Figure 2F,G). These findings suggest that periodontitis during orthodontic treatment might result in a slight loss of bone mass and may inhibit the movement of orthodontic teeth.

### 3.3. The Periodontal Microenvironment Is an Important Factor in Determining the Rate of OTM

The specimens were next sectioned and stained to evaluate the reasons for the differences in the rates of OTM among the four groups. New bone formation and osteoclasts numbers were observed using the hematoxylin and eosin, Masson, and tartrate-resistant acid phosphatase staining methods. The results showed that the amount of new bone and the number of osteoclasts in the O group were significantly higher than those in the other groups. The immunohistochemistry results also showed that the expression level of the osteogenic marker Col1a1 in the O group was higher than those in the other three groups; moreover, the inflammatory markers TNF-α and IL6 were highly expressed in the OP and OPL groups (Figure 3A and Appendix A). The enzyme-linked immunosorbent assay showed that the bone metabolism indices in the O group were higher than those in the OP and OPL groups. Taken together, these results indicated that the OTM microenvironment in the O group was more suitable for bone remodeling (Figure 3B and Appendix A).

### 3.4. CD301b+ Macrophages Are Closely Related to the Efficiency of the OTM

To further explain the differences in orthodontic effects in different microenvironments, we focused on the CD301b+ macrophage subset. The immunofluorescence staining of the specimens showed that the number of CD301b+ macrophages in the O group was significantly higher than in the other three groups (Figure 4A). Likewise, the flow cytometry results showed that the proportion of CD301b+ macrophages in the gingival soft tissues around the orthodontic mobile teeth (first molars) was significantly higher in the O group when compared to the other three groups (Figure 4B,C and Appendix A). In order to further determine the relationship between CD301b+ macrophages and the efficiency of the OTM, orthodontic models were established after the periodontitis was relieved for 3, 7, 14, and 21 days. Flow cytometry was used to detect the number of CD301b+ macrophages in the peri-gingival tissues in each group (Figure 4D and Appendix A). The results showed that the number of CD301b+ macrophages in the periodontal tissue was highest on the 14th day of delineation and returned to normal levels on the 21st day. Furthermore, micro-CT was performed to compare the extent of movement and bone parameters in each group (Figure 4E–H). The longest distance of tooth movement was observed after 14 days of delineation. The BV/TV analysis showed the highest values in the 7D and 14D groups, while there was no significant difference in trabecular bone data between these two groups. These findings indicated that CD301b+ macrophages might be beneficial for OTM.

### 3.5. CD301b+ Macrophages Can Positively Regulate the Rate of OTM and the Periodontal Microenvironment

The above studies have shown that CD301b+ macrophages are closely related to tooth movement. Therefore, we next examined whether CD301b+ macrophages contribute to tooth movement by altering the periodontitis microenvironment. CD301b+ and CD301b- macrophages were sorted from primary macrophages via flow cytometry and reinfused into the periodontium on both sides of the orthodontically treated teeth in an orthodontic rat model with mild periodontitis (Figure 5A,B). After 10 days of orthodontic treatment, the distance of the movement of the teeth in the CD301b+ macrophage group was significantly greater than that in the CD301b- macrophage group (Figure 5C,D); additionally, reduced alveolar bone resorption (Figure 5E,F) and improved periodontal conditions were observed (Figure 5G,H) in the CD301b+ macrophage group.

### 3.6. CD301b+ Macrophages Can Promote Osteogenesis and Enhance Bone Remodeling during Orthodontic Treatment

To determine how CD301b+ macrophages alter the periodontitis microenvironment, thereby enhancing the efficiency of OTM, we collected the culture supernatants of CD301b+/CD301b- macrophages and used them to stimulate bone marrow stromal cells (BMSCs). The BMSCs cultured with CD301b+/CD301b- macrophage culture supernatant were used as the experimental group, and the BMSCs cultured with normal medium were used as the control group. The cell scratch and CCK8 assays were used, and the supernatants of CD301b+ macrophages were found to significantly promote the migration and proliferation of BMSCs (Figure 6A–C). To further mimic the in vivo orthodontic environment, we applied an appropriate amount of stretching force to the BMSCs (Figure 6D), followed by the addition of the supernatants of the CD301b+/CD301b- macrophage culture supernatants. The alkaline phosphatase and alizarin red staining of the BMSCs revealed that the CD301b+ macrophages were more effective in promoting osteogenesis (Figure 6E). The quantitative real-time polymerase chain reaction (qRT-PCR) showed that the supernatants of CD301b+ macrophages induced the expression of osteogenic-differentiation-related indicators such as Runx2, Ocn, Opn, and Col1a1 in the BMSCs, which suggested that these cells might contribute to osteogenic differentiation in orthodontic environments (Figure 6F and Appendix A).

## 4. Discussion

Among the many immune cells in the body, macrophages have received extensive attention in terms of the physiological and pathological processes that occur within the body owing to their complex functions. In this study, a single-cell sequencing analysis revealed the role of macrophages in regulating OTM and the microenvironment during orthodontic treatment. Subsequently, two models of OTM in periodontitis and normal settings were created for comparison; the macrophages were polarized toward the conventional classification M1, resulting in diminished OTM distances, as reported previously [27]. Furthermore, in a mildly inflamed environment, the mechanical forces were found to slow down osteogenesis, osteoblast formation, and bone metabolism, which are detrimental to OTM.

At present, many scientists want to change the environment of the disease through a certain cell population, so as to achieve the purpose of curing the disease [7]. Macrophages have a wide range of functions such as innate defense, foreign body recognition, phagocytosis, inflammation regulation, and tissue repair [28,29,30]. Unlike the traditional classification of macrophages (M1 and M2) [31], a group of macrophages with high expression levels of CD301b focused on regulating the immune and regenerative functions has been recently identified [18,20]. In the present study, these cells were closely related to OTM. CD301b+ macrophages positively regulate the rate of OTM and the periodontal microenvironment; additionally, they promote osteogenesis in orthodontic environments, thereby altering bone metabolism and enhancing bone remodeling.

Alveolar bone remodeling during OTM is a highly regulated process; it is a combination of local reversible damage to periodontal tissues and restoration under the influence of orthodontic forces [32] Alveolar bone remodeling alters the periodontal microenvironment and coordinates bone resorption by osteoclasts and new bone formation by osteoblasts [33]. First, a cavity is formed via bone resorption on the compressed side of the mobile tooth periodontium, followed by new bone formation within the cavity; on the tension side, mainly distraction osteogenesis is observed [34]. Bone resorption and new bone formation around mobile teeth contribute to the movement of the tooth under orthodontic forces. Macrophages are a heterogeneous group of cells, and significant phenotypic and functional differences have been observed among macrophages within the same tissue and among those in different tissues [35,36,37]. Activated macrophages produce various cytokines that can regulate or stimulate osteoclasts or osteoblasts to mediate the occurrence of associated bone loss or bone regeneration [31].

During orthodontic treatment, periodic orthodontic forces are applied to the teeth and transmitted to the periodontal tissues. The periodontal cells release mediators that alter the local bioenvironmental signals in the periodontium [38,39,40], which can interact with the immune system to promote the polarization of the macrophages. Polarized macrophages are involved in the biological response to OTM, which ultimately prepares the tooth to move to a new position during periodontal restoration by remodeling the alveolar bone. In the present study, the effects of the two types of macrophages varied with the sites during OTM and the different stages during the treatment process. Type-M1-like macrophages in the traditional classification promote inflammation during tooth movement via the expression of TNF-α and inducible nitric oxide synthase (iNOS) [41], whereas type-M2-like macrophages play an important role in angiogenesis, anti-inflammatory factor secretion, and the promotion of tissue repair regeneration and wound healing [42,43]. For orthodontic treatments in inflammatory environment, the traditional classification of macrophages, as M1-like or M2-like, has advantages and disadvantages for bone reconstruction during orthodontic treatment. This classification does not strictly represent a macrophage phenotype that can modulate the microenvironment and promote OTM in a mildly inflammatory setting. If a large number of macrophages of a certain phenotype are imported to regulate bone remodeling, promote tooth movement, and enhance orthodontic efficiency, they can cause an imbalance in bone metabolism, which may result in a series of adverse effects, such as root resorption. Therefore, it is difficult to be a new target for orthodontic treatment under this type of macrophage typing in mild inflammatory environments.

In contrast, the new classification of macrophages, which is distinguished from the traditional classification, was used to describe the phenotype of macrophages from different perspectives in this study. To some extent, the obtained subsets overlapped rather than remained independent. The CD301b+ macrophages in the new classification contain a partial M1-like macrophage subset and a partial M2-like macrophage subset. Therefore, these macrophages possess the activities of both the M1-like (regulation of osteolysis and immunity) [44,45,46] and M2-like (regenerative repair and osteogenesis) macrophages [47,48,49]. In mildly inflammatory microenvironments, they respond positively to mechanical forces, enhance bone metabolism, and promote bone remodeling, thereby shifting the microenvironment towards a direction suitable for OTM. The results of the present study suggest that CD301b+ macrophages are tailored under the guidance of this goal.

The regulation of CD301b+ macrophages in bone remodeling under mechanical force during OTM has not been thoroughly studied. One of the limitations of this study is that the CD301b+ macrophages were obtained by sorting bone-marrow-derived macrophages. Thus, additional in-depth studies on the construction and functional validation of this fraction of macrophages and the corresponding condition-deficient rats with regenerated CD301b+ macrophages will have broader applications and extensive research values.

This study is an innovative exploration of the mechanism by which CD301b+ macrophages regulate OTM in an inflammatory environment. To the best of our knowledge, this is the first study to show that CD301b+ macrophages play an important role in OTM by promoting osteogenic differentiation. Macrophages with high expression levels of CD301b protein positively regulated the efficiency of OTM in a mildly inflammatory environment and the microenvironment during orthodontic treatment. This study provides a new idea for orthodontic treatment in inflammatory environments. However, more research is needed to explore the mechanism.

## Figures and Tables

**Figure 1 cells-12-00135-f001:**
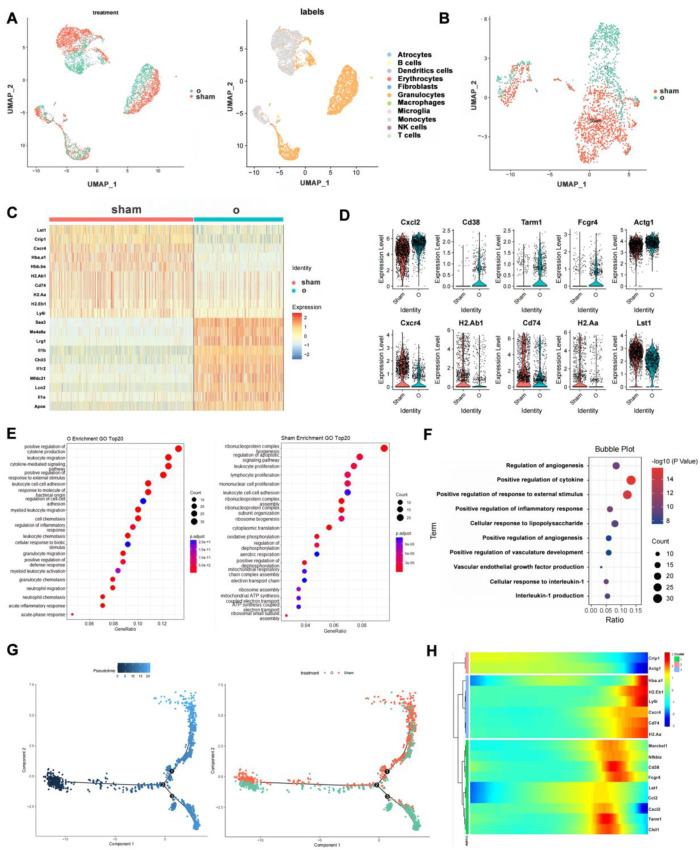
Single−cell sequencing data indicating that the macrophages in orthodontic alveolar bone have a positive effect on orthodontic tooth movement. Publicly available scRNAseq datasets were obtained from the Gene Expression Omnibus DataSets (GSM5640072 and GSM5640073): (**A**) uniform manifold approximation and projection (UMAP) plots of the total CD11b+ immune cell population isolated from the alveolar bone of normal and orthodontic mice; (**B**) UMAP plots of macrophages only; (**C**) Heat map of differentially expressed genes (DEGs) by two groups of macrophages, showing the top 10 DEGs; (**D**) Violin plot of DEGs for each group of macrophages; (**E**) Bubble plots showing the enrichment of GOBP terms in the two groups of macrophages; (**F**) Bubble plots of GOBP terms for DEGs in orthotopic groups; (**G**) Pseudotime and single−cell trajectory analyses using Monocle; (**H**) Heat map of representative pseudotime−dependent genes.

**Figure 2 cells-12-00135-f002:**
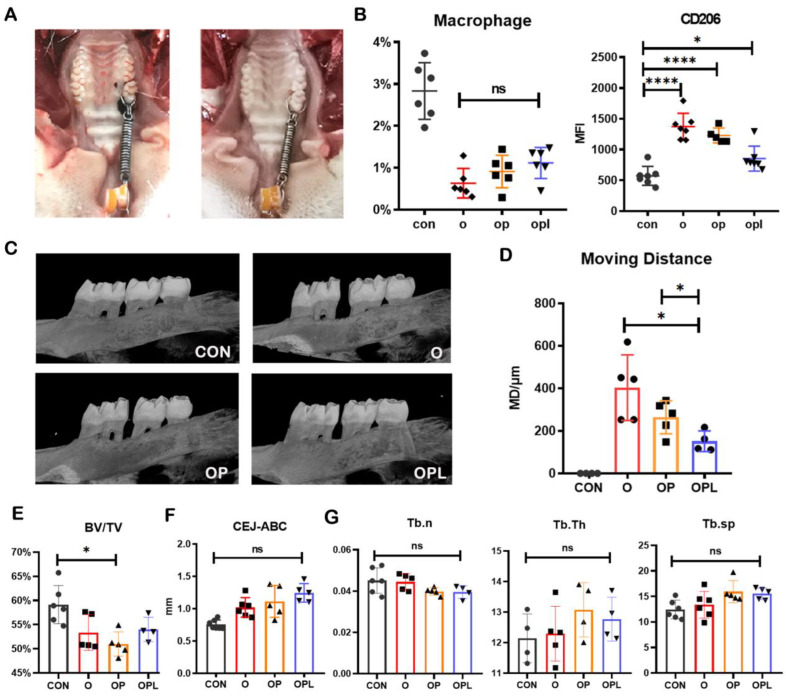
Effects of the inflammatory environment on the macrophage and orthodontic tooth movement efficiency. (**A**) The left side of the figure shows an orthodontic model in a periodontitis environment. A loop of silk thread was used at the neck of the first molar to establish the periodontitis environment. On the right is an orthodontic model in a normal environment. (**B**) Flow cytometry was used to detect the proportion of macrophages and the content of polarization markers in the periodontal tissue of each experimental group and control group. (**C**) Micro-CT images of the alveolar bone. The distances of tooth movement and the surrounding cementum are shown for each experimental group and control group. (**D**) Orthodontic tooth movement distances in each group. (**E**) Values of BV/TV for each group. (**F**) CEJ-ABC distances. (**G**) Various parameters of bone trabeculae. Values are presented as means ± SD. Note: * *p* < 0.05, **** *p* < 0.0001, ns = no significance; *n* = 4–7.

**Figure 3 cells-12-00135-f003:**
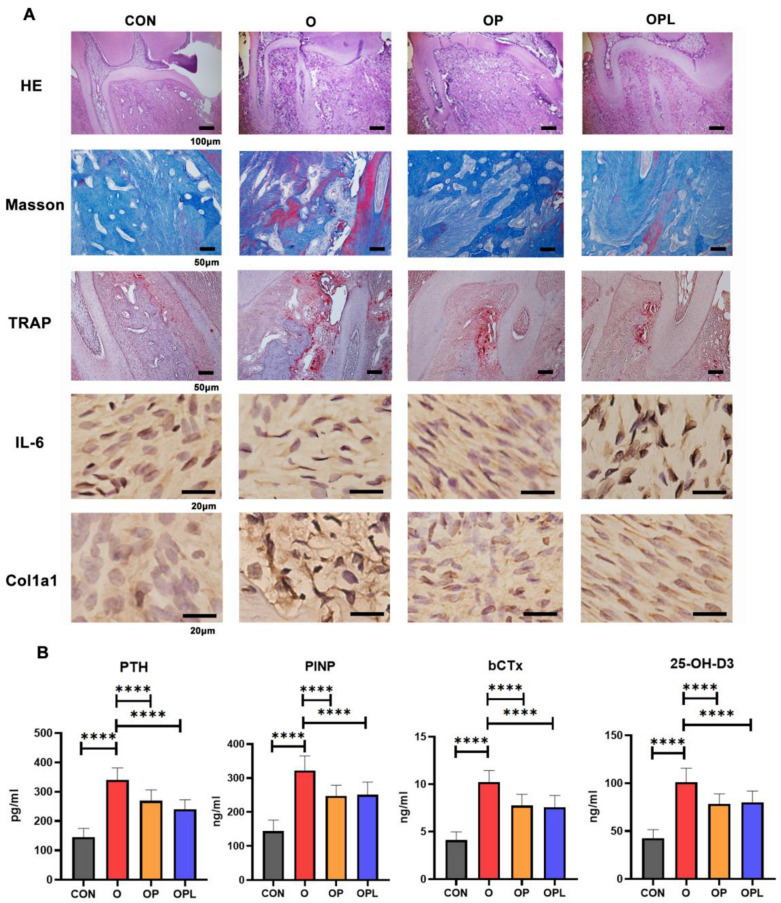
Comparison of the periodontal microenvironments of orthodontically mobile teeth in each experimental group: (**A**) representative images of H&E (scale bar = 100 μm), Masson, and trap staining (scale bar = 50 μm) methods and expression levels of immunohistochemical osteogenic and inflammatory markers (scale bar = 100 μm) in the periodontal region of orthodontically mobile teeth; (**B**) expression of bone metabolic markers. Values are presented as means ± SD. Note: **** *p* < 0.0001; *n* = 4–7.

**Figure 4 cells-12-00135-f004:**
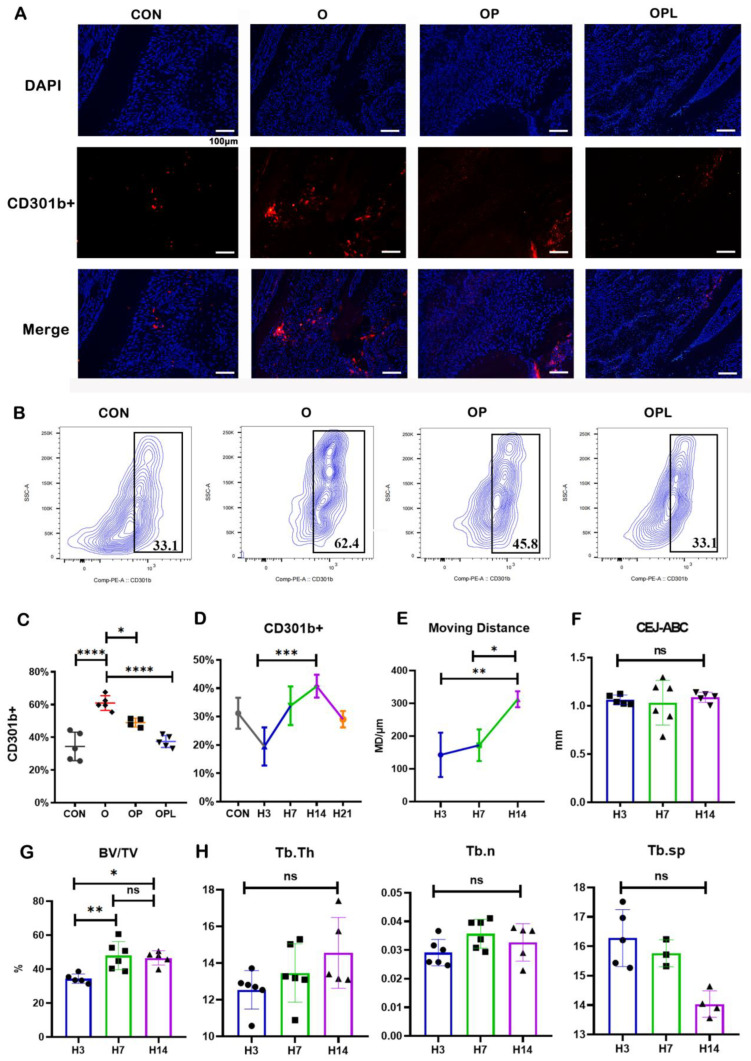
The CD301b+ macrophage content is closely related to the efficiency of the orthodontic tooth movement and the regulation of the periodontal microenvironment: (**A**) Immunofluorescence staining of CD301b+ macrophages. (**B**) Flow cytometry detection of CD301b+ macrophages. The percentage of CD301b+ macrophages of all macrophages. (**C**) An analysis of the number of CD301b+ macrophages. (**D**) Flow cytometry detection of periodontitis after 3, 7, 14, and 21 days of uncoupling of the CD301b+ macrophage contents around orthodontically mobile teeth in the orthodontic model. (**E**) distances of tooth movement. (**F**) CEJ-ABC. (**G**) BV/TV. (**H**) Each bone’s trabecular parameters. Values are presented as means ± SD. Note: * *p* < 0.05, ** *p* < 0.01, *** *p* < 0.001, **** *p* < 0.0001; ns = no significance. *n* = 4–6.

**Figure 5 cells-12-00135-f005:**
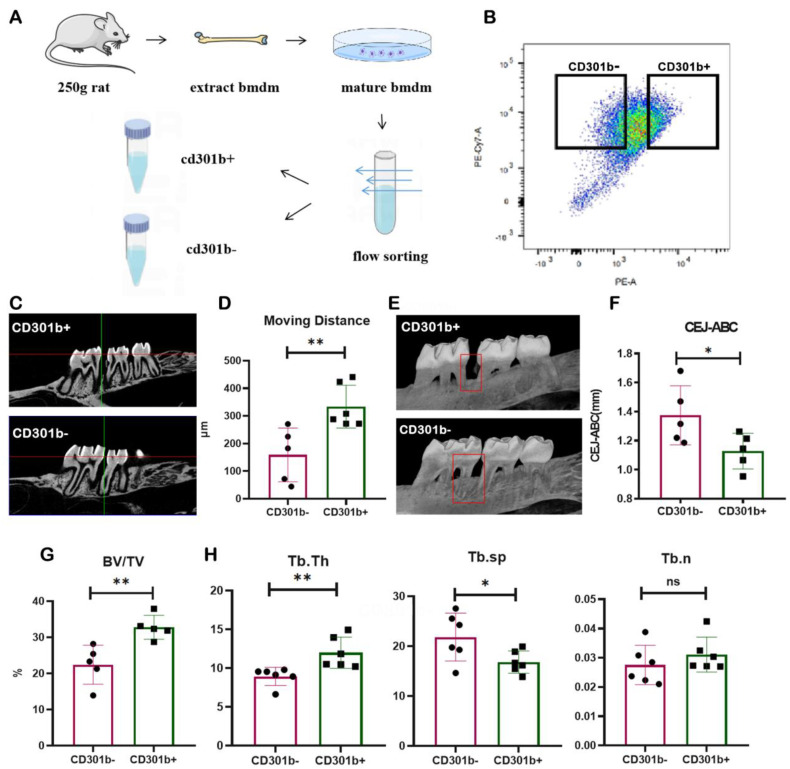
The return infusion of CD301b+ cells into the subgingival area of orthodontically mobile teeth can promote orthodontic tooth movement efficiency and improve the periodontal microenvironment: (**A**) schematic diagram of the sorted cells as well as the reinfused cells; (**B**) gating strategy diagram of cell sorting; (**C**) micro−CT image of the alveolar bone; (**D**) tooth movement distances of CD301b− and CD301b+ macrophage groups; (**E**) micro−CT image of bone resorption; (**F**) CEJ−ABC; (**G**) BV/TV; (**H**) parameters of each bone trabeculae. Values are presented as means ± SD. Note: * *p* < 0.05, ** *p* < 0.01, ns = no significance; *n* = 5−6; bar = 100 μm.

**Figure 6 cells-12-00135-f006:**
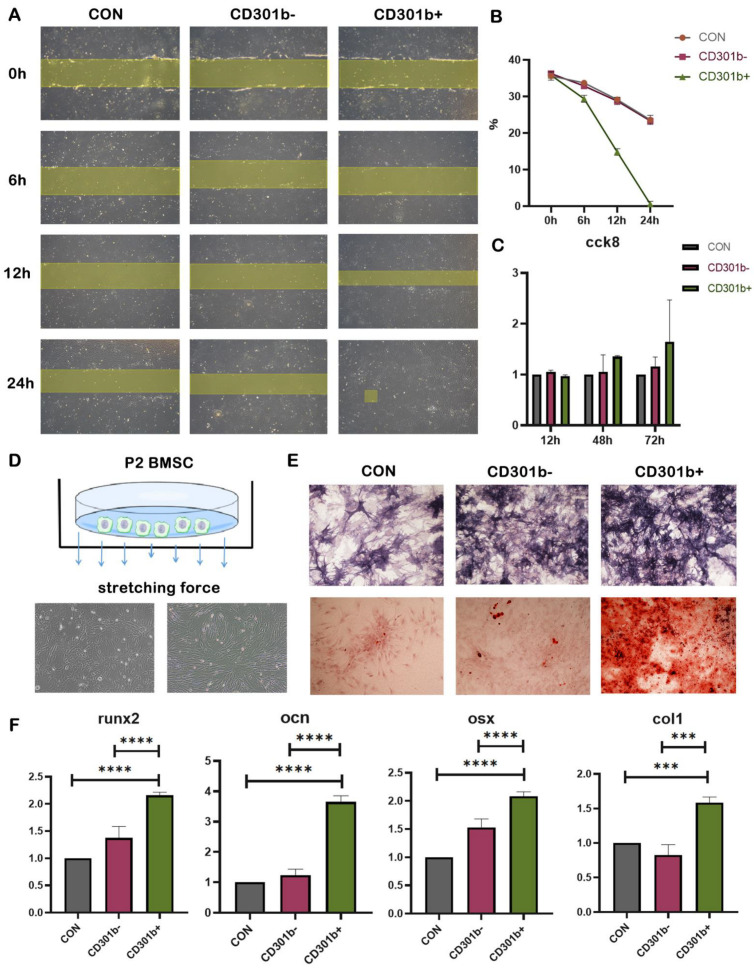
CD301b+ macrophages positively respond to mechanical forces, activate BMSCs, and promote osteogenesis of BMSCs under tension. (**A**) Cell migration assay. (**B**) Cell migration assay analysis. The area of the cell is shown as a percentage of the whole plot area. (**C**) CCK8 assay. (**D**) Morphological changes of BMSCs under tension. (**E**) ALP and alizarin red staining expression. (**F**) Levels of osteogenic-related genes (runx2, ocn, osx, col1a1) in BMSCs under tension after osteogenesis induction was detected using qRT-PCR. Values are presented as means ± SD. Note: *** *p* < 0.001, **** *p* < 0.0001; *n* = 3.

**Table 1 cells-12-00135-t001:** Flow cytometry color scheme.

Antibody	Fluorescence	Dilution Ratio
CD45	APC-cy7	1:200
CD11b	Pacific Blue	1:200
F4/80	PE-cy7	1:800
CD301b	PE	1:200
CD206	FITC	1:200

**Table 2 cells-12-00135-t002:** Primer design table for qRT-PCR.

Gene	Forward	Reverse
Col1a1	TCTGACTGGAAGAGCGGAGAG	GAGTGGGGAACACACAGGTCT
Runx2	GACTGTGGTTACCGTCATGGC	ACTTGGTTTTTCATAACAGCGGA
Ocn	GCCCTGACTGCATTCTGCCTCT	TCACCACCTTACTGCCCTCCTG
Osx	GCACAGGGTTTTGGGAGAGG	CTTGTACACCCCTGTGGACTC
Opn	CAAGGCCGAACAGGCAAAAG	TTATTGGTGGTTTGGGGGCG
Gapdh	GGCAAGTTCAACGGCACAGT	GCCAGTAGACTCCACGACAT

## Data Availability

All data generated or analyzed during this study are included in this published article and its Appendix A. All primary data supporting the results of this study can be obtained from the corresponding author upon reasonable request.

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
