# Peer review of "CD301b+ Macrophages as Potential Target to Improve Orthodontic Treatment under Mild Inflammation"

_cells, 2022, doi:10.3390/cells12010135_

Round 1
Reviewer 1 Report
Title: CD301b+ macrophages as a potential target to improve orthodontic treatment under mild inflammation
This study by Wang et al., developed orthodontic rat model of periodontitis. By using different methods, authors found that under inflammation CD301b+ macrophages promote osteogenesis, bone remodeling and orthodontic tooth movement.
I appreciate authors for this interesting study. The manuscript (MS) is nicely written and presented.
There are few minor concerns that may help to improve the MS.
-Authors used M1/M2 macrophage nomenclature throughout the MS. At the same point also mentioned that the M1/M2 is not suitable for in vivo setting. Its confusing what is authors take in this regard? Consider them calling ‘M1-like/M2-like macrophages’
Materials and methods:
Line 97-Did authors adopt any liquid/gel-based food supplement? Was the food intake normal after the surgery? What was the body weight throughout the experiment?
Line 127-It is not clear which specimen
Line 165- Is it CD45 PE-Cy7 or CD45-APC-Cy7
Line 166- Please describe CD11b fluorochrome clearly
Line 201- Its not clear how authors collected CD301b+ macrophages, are they directly from BM or cultured and stimulate to transdifferentiate? Please clarify
Results:
Line 260: Using a publicly available data sets to get some insights is a good approach. However, for the reader’s perspective, If I look at the results sections, I get the impression that authors performed single-cell sequencing. Although authors described in the materials and methods section about the data sets they used, I suggest also mentioning them in Figure 1 legend.
Line 269- The description of the rats grouping is informative.
Figure 2: Figure legends are not properly described in the overall MS.
2A-what is the difference between left and right images? B. what is ‘y’ axis? Percentage of CD45+ or CD11b+?
Macrophages (resident or recruited) numbers raises under any inflammatory conditions. Why the macrophage (%) is lower in group O, OP and OPL than control? How do authors argue this observation?
Figure 3: Quantification of histology analysis is appreciable. Please check the scale bar differences between top 3 panels. B: Description about the method is missing, please add details in the materials and methods section. Figure 4: B, C, D: Y axis % of what? Please provide gating strategy.
Figure 5: B-How authors sure about the purity of the two cell populations?
Figure 6: What is ‘control’ in this experiment
Major concern: It was reported that dendritic cells (DCs) also express the CD301b marker. Authors FACS panel did not exclude monocytes and DCs. However, since authors have tissue blocks for histology, a simple staining with F4/80, CD11c+ MHCII co-stain with CD301b can answer my concern.
Reviewer 2 Report
Dear,
A uthors investigated the role of macrophages in orthodontic treatment under inflammatory environment and they found that CD301b+ macrophages may play an active role in orthodontic treatment in an inflammatory environment and may serve as potential regulatory targets. manuscript needs minor revision. Follow the comments: introduction needs to be improved and it would be better to use the following references; https://doi.org/10.1155/2022/5304860, method and materials parts are ok, result part is ok, discussion part needs to be improved, please find the their similar articles of the mentioned above references and discussed in discussion part.
This is an interesting article and can be accepted after minor revision.
Author Response
Reviewer #2:
We are very grateful for your valuable suggestions. In response to your comments, we have revised the introduction and discussion sections of the manuscript and added relevant references. Your suggestions are essential to our research, and we would like to thank you again!
Round 2
Reviewer 2 Report
Dear,
I think, it is acceptable.